

# Fossil Sirenia from the Pleistocene of Qatar: new questions about the antiquity of sea cows in the Gulf Region

Nicholas D. Pyenson[1,2], Mehsin Al-Ansi[3], Clare M. Fieseler[1], Khalid Hassan Al Jaber[4], Katherine D. Klim[1], Jacques LeBlanc[5], Ahmad Mujthaba Dheen Mohamed[3], Ismail Al-Shaikh[6] and Christopher D. Marshall[7,8]

[1] Department of Paleobiology, Smithsonian Institution, Washington, District of Columbia, United States
[2] Department of Paleontology and Geology, Burke Museum of Natural History and Culture, Seattle, Washington State, United States
[3] Department of Biological and Environmental Sciences, Qatar University, Doha, Qatar
[4] National Museum of Qatar, Qatar Museums, Doha, Qatar
[5] Calgary, Alberta, Canada
[6] ExxonMobil Research Qatar, Doha, Qatar
[7] Department of Marine Biology, Texas A&M University-Galveston, Galveston, Texas, United States
[8] Department of Ecology and Conservation Biology, Texas A&M University, College Station, Texas, United States

Corresponding author
Nicholas D. Pyenson,
pyensonn@si.edu

## ABSTRACT

One of the largest and least documented populations of dugongs (*Dugong dugon*) resides in the coastal waters of the United Arab Emirates, and waters surrounding Saudi Arabia, Bahrain, and Qatar. The archaeological record of dugongs in the Gulf Region is abundant, but little is known about their fossil record in the region. Here we report an isolated sirenian rib fragment from the Futaisi Member of the Fuwayrit Formation near the town of Al Ruwais, in northern Qatar. The Fuwayrit Formation is a marine Pleistocene deposit exposed onshore in Qatar and the United Arab Emirates. Based on the correlative dating of the basal Futaisi Member with other onshore platforms, the rib fragment is approximately 125 ka. We propose that this isolated rib (likely the first rib from the right side) belongs to Dugongidae, with strong similarities to extant *Dugong*. We cannot, however, eliminate the possibility that it belongs to an extinct taxon, especially given its similarities with other fossil dugongid material from both Qatar and elsewhere in the world. Aside from reflecting the presence of Gulf seagrass communities in the Pleistocene, this occurrence also suggests that different (and potentially multiple) lineages of sirenians inhabited the Gulf Region in the geologic past.

## INTRODUCTION

The Arabian Gulf (also known as the Persian Gulf, but hereafter the Gulf Region; see *Fawzi et al., 2022*) is the most important region for dugongs (*Dugong dugon* (*Müller, 1776*)) in the western portion of the range of this species, which extends from East Africa to Oceania

(*Marsh, O'Shea & Reynolds, 2011*). The Gulf Region possesses the second largest population of dugongs in the world, distributed in coastal waters near the United Arab Emirates (UAE), Saudi Arabia, Bahrain, and Qatar. In the Gulf Region, there are three dugong distribution hotspots: the coastal area of the UAE near Marawah Island; the coastal region of Saudi Arabia between the UAE and Qatar; and the northwest coast of Qatar from the Zekreet Peninsula and the Hawar Islands to Ras Ushayriq and offshore to Fasht Adhm, Bahrain (*Preen, 1989*; *Marsh et al., 2002*; *Preen, 2004*; *Marshall et al., 2018*). However, this entire population is vulnerable to exploitation and it is listed as Vulnerable to Extinction by the International Union for Conservation of Nature (*Marsh & Sobtzick, 2015*). Historically, dugongs have had a cultural and economic importance in the Gulf Region since the Neolithic period, approximately 7,500 years ago (*Méry et al., 2009*; *Beech, 2010*), although little is known about the ecology and population dynamics of the Gulf Region's dugong population today. Recent fieldwork in western and northern Qatar (*Marshall et al., 2018*) has provided the first steps towards better documenting this vulnerable species, which lives alongside major petroleum development sites in the Gulf.

Nearshore marine deposits throughout the Gulf Region (*Al-Saad & Ibrahim, 2002*; *Dill et al., 2005*) provide abundant preservation potential for fossil marine mammals in the region, yet the published record is limited to preliminary reports by *LeBlanc (2009, 2021)*. *LeBlanc (2009, 2021)* indicated extensive fossil material throughout Qatar, including fossil marine vertebrates from Eocene to Miocene in age. In particular, *LeBlanc (2009, 2021)* noted abundant fossil Dugongidae from early Miocene (Aquitanian-Burdigalian) localities in the Dam Formation of southwest Qatar.

Outside of Arabia, fossil Dugongidae are well known from localities in the western margins of the former Tethys Sea, including Eocene-Oligocene nearshore deposits of Egypt in the Fayum Basin and the Eocene of Europe, representing lineages of completely aquatic dugongids that are distantly related to today's *Dugong*. Fossil Dugongidae from the eastern Tethys are less well known, but provide striking evidence for multispecies sirenian communities that occur globally during the Cenozoic (*Vélez-Juarbe, Domning & Pyenson, 2012*). Fossil dugongids from the early Miocene of western India include an assemblage of dugongine dugonids belonging to the genera *Kutchisiren Baipai et al. (2010)*, *Bharatisiren Bajpai & Domning (1997)*, and *Domningia Thewissen & Bajpai (2009)*, which co-occurred in early Miocene (Aquitanian) Khari Nadi Formation (*Baipai et al., 2010*), with some lineages, such as *Bharatisiren*, ranging into late Oligocene rocks of India. Beyond the Tethys, the Kutch assemblage is complemented by another multispecies sirenian assemblage from marine beds of Nosy Makamby on the island of Madagascar (*Samonds et al., 2019*). While the stratigraphic constraints on the age of this assemblage are broad (from early to late Miocene), the presence of four dugongids from Nosy Makaby includes three dugongines and one halitherine: *Rytiodus heali Domning & Sorbi (2011)*, *Norosiren zazavavindrano Samonds et al. (2019)*, an unidentified dugongine, and *Metaxytherium* cf. *M. krahuletzi Depéret (1895)*. Dugongids are also represented in Eocene deposits on Madagascar by *Eotheroides lambondrano Samonds et al. (2009)*. Other dugongid material from Oceania are sparse and represent isolated material (*Fitzgerald, 2005*; *Pledge, 2006*; *Fitzgerald, Vélez-Juarbe & Wells, 2013*).

In light of these western and eastern Tethys assemblages, sirenian material from the Middle East provide insight in the geographic distribution and occurrence of taxa that comprise these multispecies communities. Building on preliminary fieldwork from 2018 to 2020, we here report the northern-most Qatari record of fossil Dugongidae based on an isolated rib collected from the Pleistocene Futaisi Member of the Fuwayrit Formation, exposed near the town of Al Ruwais in northern Qatar. These fossil data provide an important comparison to the modern record, showing that a lineage of ecologically important marine plant consumers in Qatar extends at least into the Pleistocene. Our finding fits with the presence of other Burdigalian fossil Dugongidae in Qatar, providing a nearly 20-million-year history of sea cows in waters surrounding Qatar. This timeframe includes a period prior to the Pliocene origin of the Gulf and the orogeny of the Zagros mountain belt; this open communication between the eastern and western Tethys may have permitted multiple species of dugongids to survive into the late Neogene of Arabia.

## MATERIALS AND METHODS

**Museum Abbreviations.** QM, QNM: National Museum of Qatar, Qatar Museum Authority, Doha, State of Qatar; UM and GSP-UM, University of Michigan Museum of Paleontology and Geologial Survey of Pakistan-University of Michigan, Ann Arbor, Michigan, USA; USNM PAL and USNM: Departments of Paleobiology and Vertebrate Zoology, National Museum of Natural History, Smithsonian Institution, Washington, District of Columbia, USA.

**Specimens Observed.** Specimens Observed. *Callistosiren boriquensis Vélez-Juarbe & Domning, 2015* (USNM PAL 542423), *Dugong dugon* (USNM 197900, 257107, 284440, 284441, 550456), *Hydrodamalis gigas* (*Zimmermann, 1780*) (USNM 35638, 218380, 21258), *Metaxytherium calvertense Kellogg, 1966* (USNM PAL V16715), *Nanosiren sp. Domning & Aguilera, 2008* (USNM PAL 16630, 559306), *Pezosiren portelli Domning, 2001* (USNM PAL 542909, 542910), Prorastomidae cf. *Pezosiren portelli* (USNM PAL 542916), *Priscosiren atlantica Vélez-Juarbe & Domning, 2014* (USNM PAL USNM 542417), *Protosiren sattaensis Gingerich et al. 1995* (cast of GSP-UM 3001), *Protosiren smithae Domning & Gingerich, 1994* (cast of UM 94810), *Trichechus inunguis* (Natterer in von *Pelzeln, 1883*) (USNM 20916), *Trichechus manatus Linnaeus, 1758* (USNM 1375, 14334, 217259, 238018, 256674, 527899, 527901, 527902, 527904, 527905, 527906, 527907, 527909, 527910, 527916, 550462).

**Comparative morphology**. We used the aforementioned collections to compare anterior rib morphology among fossil and living sirenian taxa. We adopted rib measurements provided by *Zalmout & Gingerich (2012*: Fig. 11 and Table 9*)* in the following way: the lateral and anteroposterior thicknesses, *via* hand calipers, of the rib at the angle (at the level of the neck diameter in *Zalmout & Gingerich, 2012*); and the lateral and anteroposterior thicknesses, *via* hand calipers, of the rib at midshaft (equivalent to MAWM, maximum anteroposteriorly width of midshaft, and MLWM, maximum medolateral width at midshaft in *Zalmout & Gingerich, 2012*). We measured these distances in the associated

first and second ribs of fossil and living sirenian taxa (see Supplemental Dataset), averaging right and left sides, where such pairs existed.

## SYSTEMATIC PALEONTOLOGY

Mammalia *Linnaeus, 1758*
Sirenia *Illiger, 1811* sensu *Velez-Juarbe & Wood, 2018*
Dugongidae *Gray, 1821* sensu *Velez-Juarbe & Wood, 2018*

**Specimen**. QM.2021.0504 consists of an incomplete rib fragment belonging likely to the first right rib. QM.2021.0504 lacks the capitulum but it preserves diagnostic features belonging lacking morphology proximal from the capitulum, but it preserves diagnostic features from the level of the capitulum to the distalmost end. See sections below for more on the description, anatomical and taxonomic identity of QM.2021.0504.

**Locality.** We collected the rib fragment QM.2021.0504 on 9 December 2019 at N 26°06′44.6″, E 51°09′31.2″ outside of Al Ruwais, Baladīyat ash Shamāl, State of Qatar under the authorization of the National Museum of Qatar (Qatar Museums).

**Stratigraphy, depositional environment and age.** The modern-day peninsula of Qatar is the result of Arabian-Eurasian tectonic plate convergence in the Neogene and the Zagros orogeny in modern-day Iran during the Plio-Pleistocene. This tectonism created fold-thrust belts in Iran and a wedge-shaped, low angle foreland basin in the Gulf Region with mixed evaporites, carbonates, and siliciclastics deposited on the Arabian plate, along with several depositional hiatuses caused by Neogene sea-level change (*Sharland et al., 2004*; *Perotti et al., 2011*). The peninsula of Qatar is a wide anticlinal dome, slightly warped by the north-plunging regional Qatar-South Fars Arch (*Cavelier, 1970*; *Rivers & Larson, 2018*) and the Dukhan anticline, which is restricted to the western portion of the peninsula (Fig. 1A; *Dill, Nasir & Al-Saad, 2003*; *Al-Saad, 2005*).

The majority of surficial carbonate rocks exposed in Qatar consist of middle Eocene limestones belonging to the Rus (Ypresian) and Dammam (Lutetian) formations (*Cavelier, 1970*; *Al-Saad, 2005*). The Dammam Formation is overlain by Miocene limestones, marls, and shales belonging to the Dam Formation (Aquitanian-Langhian), and middle Miocene sands and gravels of the Hofuf formation (Messinian-Gelasian) in the southwest portion of the peninsula (Fig. 1A; *Al-Saad & Ibrahim, 2002*; *Alkhaldi, Read & Al-Tawil, 2021*; *LeBlanc, 2021*). Along the northern and northwestern coastal margins of Qatar, Pleistocene deposits belonging to the Fuwayrit Formation overlie the crystalline dolomites of the Dammam Formation, and three sequences are exposed onshore: the Futaisi Member, the Dabb'iya Member and the Al Wusayl Member (*sensu Williams & Walkden, 2002*; see also *LeBlanc, 2022*; *Williams, 1999*). Elsewhere in the Gulf, the Fuwayrit Formation is exposed onshore near Abu Dhabi in the United Arab Emirates, although only the Futaisi and Dabb'iya members are represented (*Williams & Walkden, 2002*). The Fuwayrit Formation is also the only marine Pleistocene sequence exposed onshore around the Gulf; the aeolian Ghayathi Formation and the continental Aradah Formation comprise the other two Quaternary sequences (*Williams & Walkden, 2002*).

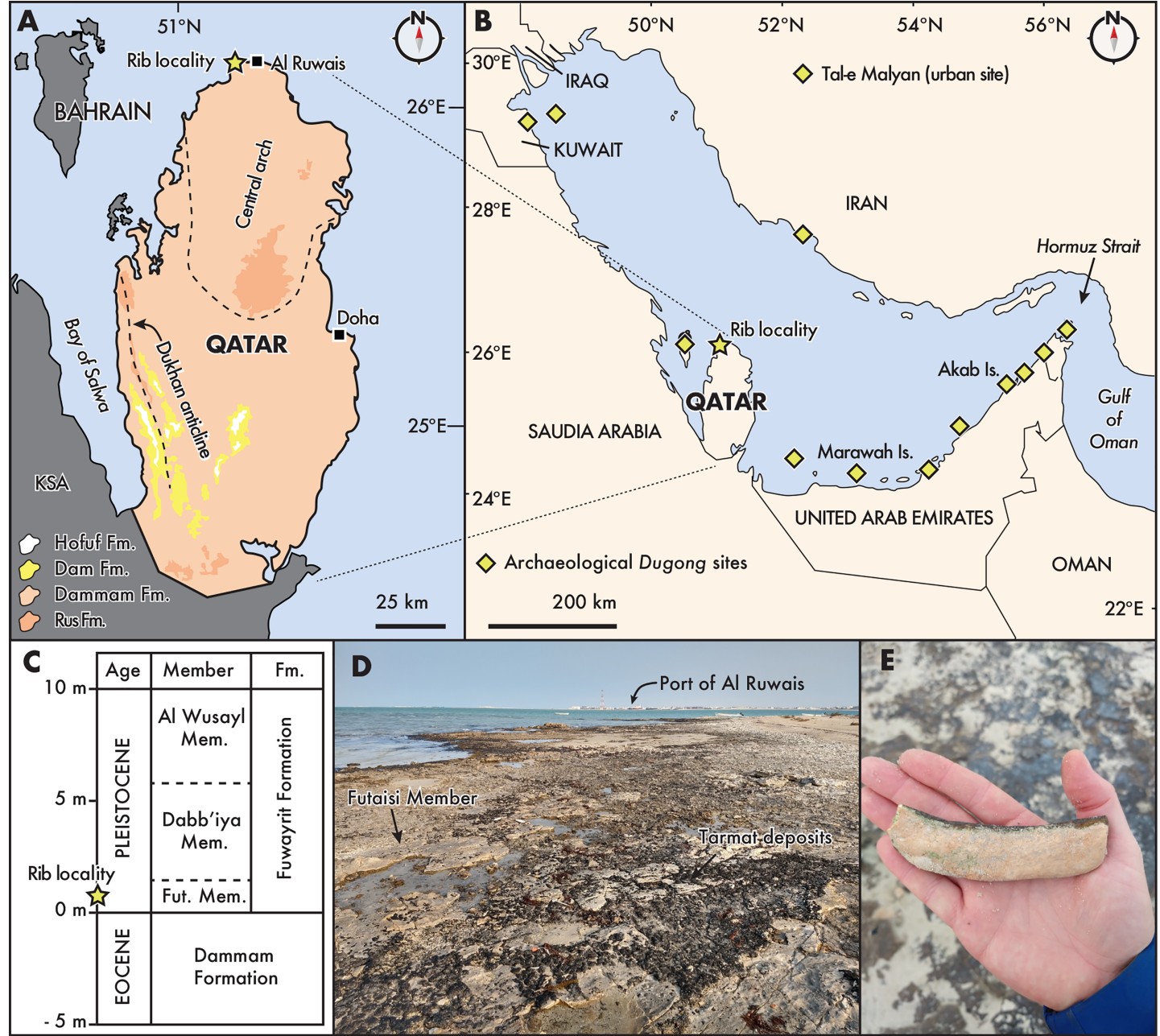

**Figure 1 Distribution and geological context of archaeological and fossil Dugongidae from the Gulf Region.** (A) State of Qatar, with the rib locality denoted by a star, near the town of Al Ruwais. Pleistocene outcrops are located along the coastal margins (see *Rivers & Larson, 2018*). Eocene (Rus and Dammam formations) and Miocene (Dam and Horfuf formations) age units after *Cavelier (1970)*, *Al-Saad (2005)*, and *Rivers & Larson (2018)*; (B) The Gulf Region, after *Hughes (1997)*. *Dugong* archaeological sites follow *Beech (2010)*; (C) Generalized stratigraphic section of onshore deposits in northern Qatar, relative to modern sea level, modified from *Williams & Walkden (2002*: Fig. 4); (D) and (E), showing the fossil locality west of the Port of Al Ruwais, with Pleistocene age outcrops of the Futaisi Member of the Fuwayrit Formation. Fut., Futaisi; KSA, Kingdom of Saudi Arabia.

The shoreline west of Al Ruwais, near the locality of QM.2021.0504, exposes the Futaisi Member, which is the earliest Pleistocene deposit preserved on the peninsula of Qatar (also see mapping in *Rivers et al., 2020*: Fig. 2). The Futaisi Member does not exceed 1 m in

thickness anywhere along its exposure, and the maximum elevation at which it is found is approximately 1.5 m above present high-tide level (Fig. 1C; *Williams, 1999*). At the locality, the deposits contain abundant pebbles of dolomite reworked from the nearby brecciated terra rossa (*i.e.*, angular, poorly sorted dolomitic clasts), which are set in a matrix of marine allochems and micrite (*Williams, 1999*: Table 12); while we cannot exclude that the rib fragment was similarly reworked from the underlying Eocene deposit, it shows only slight taphonomic signs of wear (see next section), suggesting limited transport or abrasion (*Fiorillo, 1988*). This lag deposit of pebbles is overlain by a well-sorted oolitic grainstone, which shows low-angle planar cross-bedding, and contains sparse sediment-filled shafts and tunnels, resembling *Ophiomorpha* and *Thalassinoides*-type burrows. Locally, the amount of bioturbation may increase with recognizable *Ophiomorpha*-type shafts and tunnels, mostly filled with sediment. Occasional invertebrate burrows towards the top of the unit are filled with poorly sorted coarse skeletal sand, and occasional flattened, sub-rounded dolomite pebbles, which may represent storm-filled burrows.

The Futaisi Member has been interpreted as a transgressive deposit recording only the peak of a Pleistocene high-stand (*Williams, 1999*; *Williams & Walkden, 2002*). The low-angle, planar cross-bedding, combined with the trace-fossil assemblage, suggests that deposition of the Futaisi Member occurred in a foreshore-upper shoreface environment. Also, the well-rounded, well-sorted nature of the sediments reflects a moderately high-energy (*i.e.*, wave-dominated) setting. The degree of bioturbation reflects the difference between predominantly subaqueous and predominantly subaerial settings. The most heavily burrowed sediments were frequently submerged, whereas the wavy laminated sediment was probably deposited high up the beach, and rarely flooded afterwards. The interpretation of a foreshore environment is also supported by the recognition of a grain-contact meniscate cement as the earliest cement generation in some samples.

Dating the age of Pleistocene deposits in the Gulf Region is challenging. Luminescence dating of Ghayathi Formation aeolianites from localities outside of Qatar produced age ranges between 45 and 130 ka (Upper Pleistocene, Stage 4), although these dates are likely distorted by diagenetic contamination; similar techniques on sediments from the Fuwayrit Formation have been unfruitful (*Williams, 1999*). *Williams & Walkden (2002)* reported radiocarbon dates on coral, red algae, and barnacle samples from the Dabb'iya member of the Fuwayrit Formation in Qatar and the UAE at approximately 30 ka. While diagenetic alteration cannot be excluded, the unit overlying the Futaisi member is clearly not Holocene in age.

Lacking direct age dates for the Fuwayrit Formation, *Williams & Walkden (2002)* inferred the age of these sediments by comparing their elevation and stratigraphic thickness with similar deposits from stable shoreline platforms elsewhere in the world. This approach operates on two assumptions: (1) the Fuwayrit Formation represents the youngest pre-Holocene marine deposits in the Gulf; and (2) either there is no Pleistocene tectonism along the coastline (*e.g.*, isostatic rebound), or the onshore deposits are unaffected by tectonism. Based on the elevation and stratigraphy of the Fuwayrit Formation compared with other onshore Pleistocene platforms in Bermuda, the Bahamas,

the Mediterranean, New Caledonia, Hawaii, the eastern United States, and Australia, *Williams & Walkden (2002)* argued the Fuwayrit Formation was deposited during the last interglacial (Marine Isotope Stage 5e; *Railsback et al., 2015*). In turn, this inference suggests that the basal sediments of the Futaisi Member at the locality of QM.2021.0504 are approximately 125 ka.

**Description**. The rib fragment (QM.2021.0504) is about 112 mm long in a straight line, 31 mm wide at its greatest lateral extent and 17 mm thick at its thickest anteroposterior direction. The fragment is missing its proximal-most processes, including the neck and capitulum, but it appears to preserve the distalmost termination of the rib (Figs. 2A and 2B). Along the main axis of the rib shaft, its greatest lateral extent and thickest anteroposterior dimension coincide at the same level, which is within the proximal third of the fragment, presumably the angle of the rib. In this area, the shaft exhibits a slight bowing and lateral pinching, with the proximal-most part terminating in a postmortem breakage through the neck, revealing a broadly triangular cross-section. The raised prominence, which lacks any articular facet (*i.e.*, tuberculum) but marks a change in surface topography, is likely part of the angle. Its distal surface likely provided an attachment for the serratus magnus and scalenus muscles (*Domning, 1977*, Figs. 41 and 42; see anatomical identity, below in the Results & Discussion). The breakage at the proximal end of the rib reveals dense inner bone architecture that is osteosclerotic (Fig. 2E). There is minimal, if any, hyperplasy on the periosteal cortices of the rib fragment, suggesting little to no pachyostosis (*sensu de Buffrénil et al., 2010*).

Overall, the rib fragment's axis is lightly curved and the rib is subcylindrical in cross-section, with distal edges that are rounded. In anterior and posterior view, the fragment is gently curved, with a slight twisting from its main axis in the proximal-most third (Figs. 2E and 2F). The distal end is tapered, with rounded edges and square termination in anterior and posterior views. The bone surface shows light cracking, and it is slightly abraded and weathered, displaying weathering Stage 1 for both these latter categories according to *Behrensmeyer (1978)*. There is no rectangular flaking on the bone surface, nor any sharp breaks in the bone. The external surface shows no evident crab traces (see *Pyenson et al., 2014*), yet there appear to be scavenging marks on the bone surface from an unknown tracemaker (Fig. 2D; *Higgs & Pokines, 2014*). The anterior surface is covered in tar residue (Figs. 2B and 2C) that is abundant along the northern beaches of Qatar, especially in tarmat deposits from nearby Ras Rakan Island, about 10 km northeast of the rib locality (*Arekhi et al., 2020*).

## RESULTS AND DISCUSSION

**Anatomical and taxonomic identity.** This rib fragment is diagnostically sirenian based on its subcylindrical cross-sectional profile, its slight overall curvature along its main axis, its lack of acute crests, and the presence of osteosclerosis, a trait in common with nearly all post-Eocene sirenians (*de Buffrénil et al., 2010*), except for *Dugong* and *Callistosiren* (see more about taxonomic identity below; *de Buffrénil et al., 2010*; *Vélez-Juarbe & Domning, 2015*). These features of the rib exclude other potential mammalian candidates, including

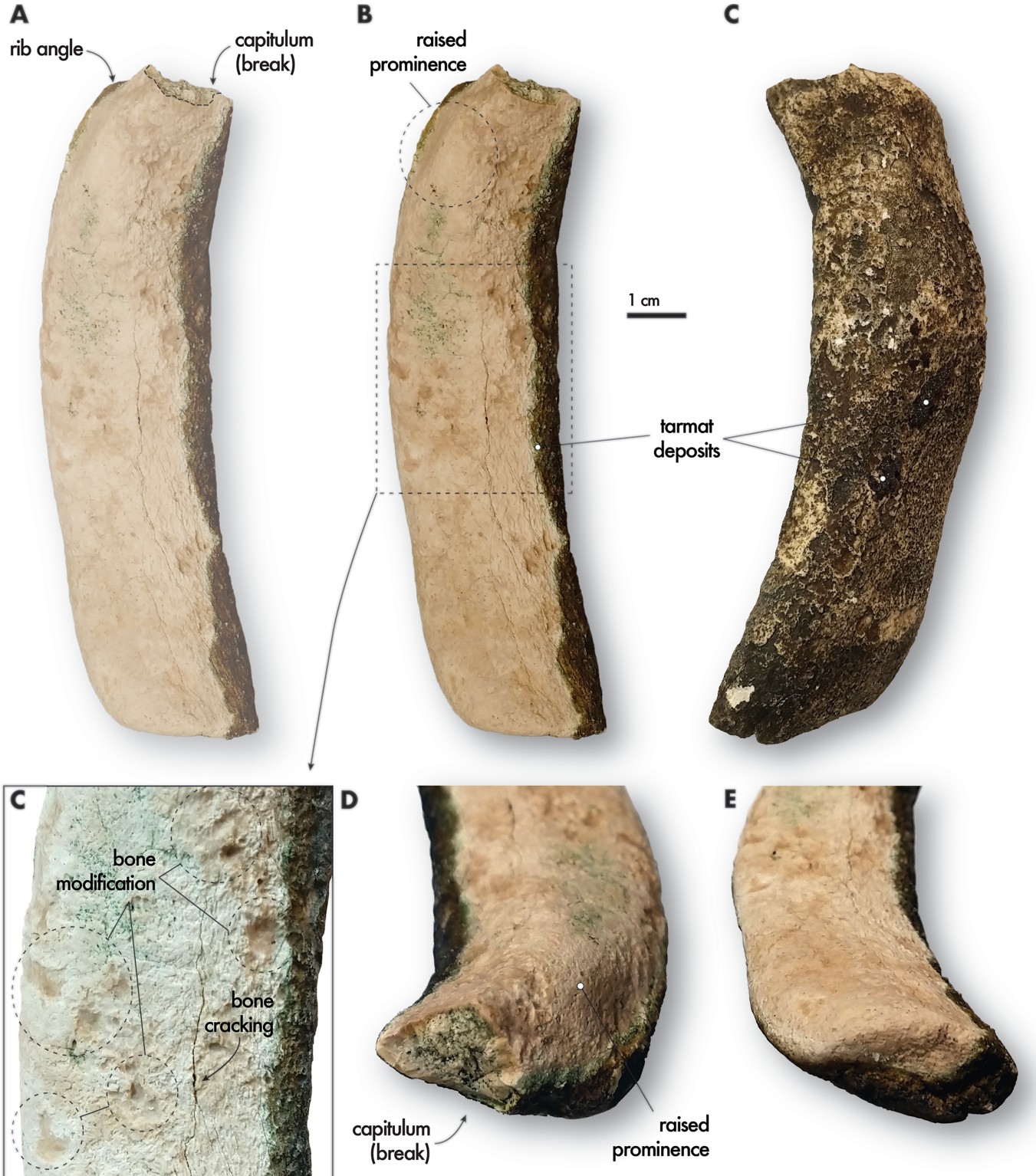

**Figure 2 Fossil dugongid rib fragment collected near Al Ruwais, Qatar.** QM.2021.0504, Dugongidae right first (?) rib in anterior (A, B) and posterior (C) views, with a close-up on the bone surface (D) showing putative scavenging traces. Oblique proximal (D) and distal (E) views of the rib fragment. Tarmat deposits are also highlighted.

terrestrial mammals, which are compressed and lozenge-shaped in cross-sectional profile, as well as marine mammals such as cetaceans and pinnipeds, whose ribs are either more quadrangular or concave in cross-section.

In the sirenian axial skeleton, this fragment most closely matches the morphology of anterior ribs (*Kaiser, 1974*). QM.2021.0504 does not belong to any of the mid-thoracic or posterior ribs because the fragment shaft axes exhibit no proximal dorsoventral flattening nor twisting near the angle; instead, it has strong anteroposterior flattening, typical of anterior ribs. The distal end appears slightly worn but well-ossified and has no clear breaks, suggesting that it represents the true distal end of the rib, although it does not exhibit any spongy surface typical of cartilaginous articulation (Fig. 2F). The raised prominence near the angle belongs on the anterior surface of the rib, which diagnoses the rib as belonging to the right side (see *Domning, 1977*: Figs. 35, 41 and 42).

The proximal breakage on the Qatari rib fragment is largely in line with the raised prominence, suggesting a closest match to the first and second ribs; this suggestion is reinforced by the general anteroposterior flattening of the shaft and its limited curvature and lack of proximal twisting nor dorsoventral flattening along its shaft axis. The rib fragment also lacks proximal tubercular facets and a relatively large neck cross-section, which all typify ribs in the mid-thoracic series (Fig. 2).

As for its taxonomic identity, the rib fragment does not belong to *Hydrodamalis*, based primarily on its much smaller size, diminished overall curvature. The first, second and third ribs of *Hydrodamalis* are strongly curved, with the first rib nearly semi-lunate in overall arc. Also, the first rib in *Hydrodamalis* presents a medial process on the internal margin of the first rib, which may have been present in QM.2021.0504 prior to the proximal breakage at the level of the neck. If the breakage indicates an approximate cross-section of the relative proportions of the neck width, this measure relative to the rib's width at the angle also precludes the second rib of *Hydrodamalis* as a potential candidate, aside from size. We also exclude *Trichechus* from potential candidates, based on two features, which are both absent in the Qatari specimen: a patent and rounded crest that extends laterally from the neck along the midline of the medial surface of the first and second ribs in *Trichechus*; and overall swelling (*i.e.*, cortical hyperplasy from pachyostosis) in the proximal half of the rib, which is especially visible on the external surface, and increases posteriorly from the first rib throughout the series.

Among extant sirenians, the Qatari specimen is most similar to the first and second ribs of *Dugong*, based primarily on the anteroposterior flattening of the rib shaft and the morphology of the raised prominence, which is preserved in the Qatari specimen. In the first and second ribs of *Dugong*, the raised prominence marks a boundary for the changes in organization of muscle attachments: on the first rib, attachments for the scalenus and serratus magnus muscles are located distally on the dorsolateral edge, while proximally the neck and head of the rib anchor the longus capitis muscle medially, and the longissimus dorsi and iliocostalis thoracis muscles laterally; on the second rib, the prominence continues providing an attachment for the scalenus and serratus muscles distally, and the longissimus dorsi and iliocostalis thoracis proximally (see Fig. S1). The distalmost tip of

the Qatari specimen shares the most similarities with the second rib in *Dugong*, with a modestly rounded rather than fluted termination (Fig. 2E).

For fossil dugongids, there are few taxa with associated first and/or second ribs; notably, no such postcrania have been clearly identified in association with diagnostic crania for eastern Tethys sirenians, which limits comparisons to taxa from the western Tethys Sea and the Americas. Among those fossil taxa with comparable ribs, the Qatari specimen is most similar to the first rib of *Callistosiren boriquensis Vélez-Juarbe & Domning (2015*: Fig. 10), although QM.2021.0504 does not preserve morphology of the neck that would be more diagnostic. QM.2021.0504 is broadly similar in morphology to the first ribs of dugongids such as *Metaxytherium* spp. (*Vélez-Juarbe & Domning, 2014*); the distal termination of QM.2021.0504 is not obviously expanded, as the first rib of *Protosiren Abel (1907)* from the Eocene (Lutetian-Priabonian) of both Egypt and Pakistan (*Zalmout, Ul-Haq & Gingerich, 2003*: Fig. 3). The overall dimensions of the rib fragment indicate that it belonged to a relatively small sirenian: much smaller than *Metaxytherium calvertense*, but larger than *Nanosiren garciae*, and most similar in size to *Priscosiren atlantica*. In our view, the preponderance of comparisons points to QM.2021.0504 belonging to Dugongidae; among described dugongid taxa, it shares the most similarities with the first rib of *Dugong*. The limited amount of preserved morphology, however, means that we cannot exclude the possibility of an unidentified, extinct dugongid.

To better characterize the morphology of QM.2021.0504, we collected measurements from a large sample size ($n = 10$ taxa; 53 specimens) of the first and second ribs of fossil and extant sirenians, including living and historically extinct sirenians (*e.g.*, *Dugong*, *Hydrodamalis*, and *Trichechus* spp.), and any relevant fossil taxa, including fossil dugongids (*e.g.*, *Nanosiren* and *Metaxytherium*), and stem sirenians such as *Pezosiren*, *Protosiren* and an unnamed taxon of prorastomid (Fig. 3). While the correlation among the first rib measurements (Figs. 3A and 3B) were higher than second rib ones (Figs. 3C and 3D), the Qatari specimen plotted closer to the regression line for measurements near the rib angle than those at midshaft. In all cases, QM.2021.0504 plotted within the variation of extant Dugong, but close to stem sirenians in the dataset. While it is possible that QM.2021.0504 represents a reworked rib from a stem sirenian originally preserved in the underling Dammam Formation, it is unlikely as one of the key diagnostic features of Eocene age sirenian first ribs is an expanded termination, which is not present in QM.2021.0504 (Fig. 2). It remains equally plausible, given these plots, that QM.2021.0504 represents an unidentified dugongid taxon.

**Evolutionary and biogeographic significance.** In the Gulf Region, marine mammals mostly have a Holocene record (*e.g.*, *Stewart et al., 2011*; see Fig. 1B); the pre-Holocene record of fossil marine mammals is sparse. *Whitmore (1987)* reported an isolated delphinoid periotic associated with fragmentary sirenian ribs from the type locality of the Dam Formation at Jabal al Lidam (*Thralls & Hasson, 1956*; *Al-Saad & Ibrahim, 2002*), about 40 km west of Dammam in Saudi Arabia. The Dam Formation also extends to southwestern Qatar, where more associated fossil dugongids have been reported by *LeBlanc (2009, 2021*; and see below), along with fragmentary cetacean remains. The size and morphology of the first and second ribs for these Aquitanian-Langhian dugongids

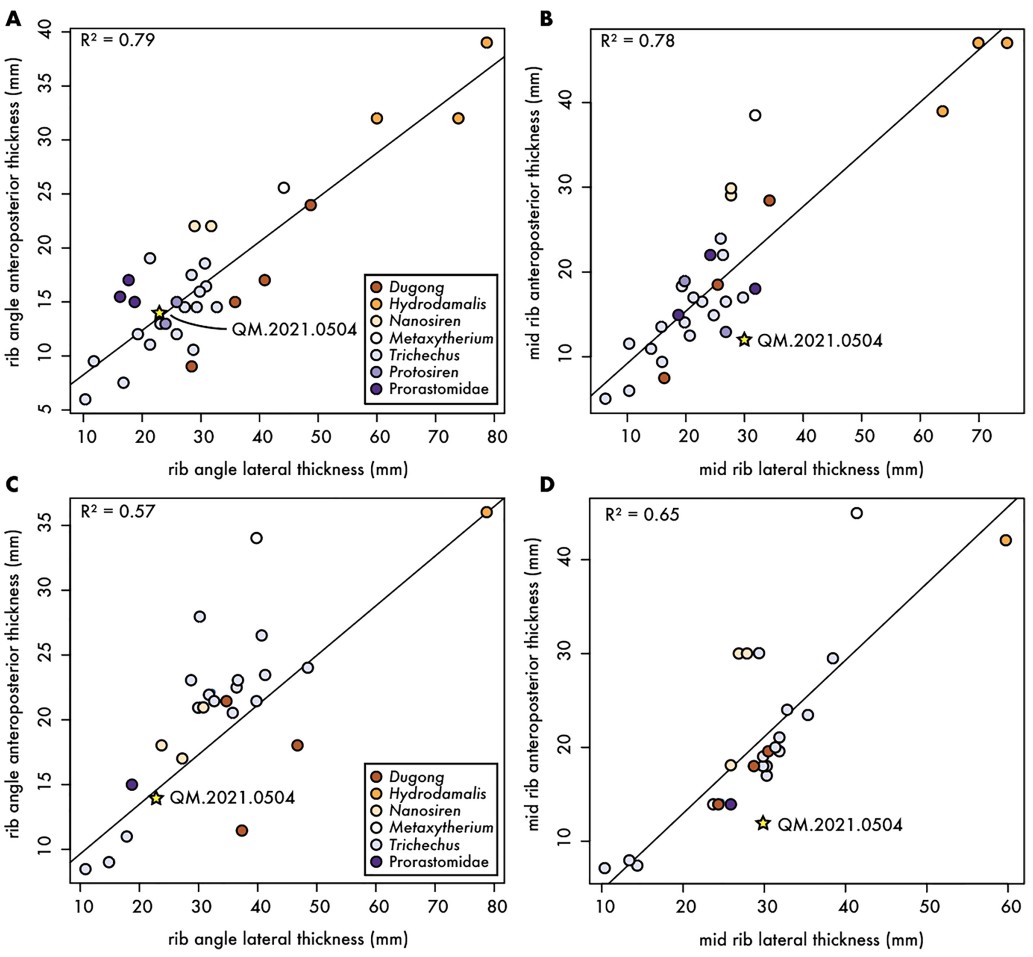

**Figure 3 Comparative measurements of first and second ribs for fossil and extant sirenians.** Anteroposterior and mediolateral dimensions of fossil and extant sirenian ribs. (A) and (B) dimensions at the rib angle and midshaft, respectively, for the first rib; and (C) and (D) dimensions at the rib angle and midshaft, respectively, for the second rib; both with $R^2$ values from linear regressions. Gold star denotes Qatari specimen reported in this article.

from southwestern Qatar (*e.g.*, QNM.2011.660.2ABC; see Article S1) is more similar to QM.2021.0504 than to extant *Dugong*, although the postcranium of dugongids has limited diagnostic use below the family level.

North of the Gulf, *Abbassi et al. (2016)* reported a dugongid skeleton from Burdigalian limestones of the Qom Formation (*Morovati et al., 2021*) near Shirin Su in central Iran, along with other sirenian fossils from elsewhere in central Iran. The reported ribs of the Shirin Su material are natural molds, which makes it difficult to compare with the Qatari material; *Abbassi et al. (2016)* were similarly limited by the preserved postcranium, and they could not assign the Shirin Su material to either the halitheriine or dugongine dugongid subfamily.

In the early Miocene, the Zagros Basin connected the Western and Eastern Tethys across the Eurasian and Arabian plates, and the current sirenian fossil record shows mostly non-overlapping taxonomic assemblages of halitheriine dugongids from the Western

Tethys (*e.g.*, halitheriines from Europe) and the Eastern Tethys (*e.g.*, dugongines from India). The Nosy Makaby dugongid assemblage from Madagascar represents a slight deviation from this pattern with three dugongines and one halitherine, although we note that the broad and unconstrained temporal range of this assemblage hinders a more precise understanding of their potential ecological co-occurence. The subsequent collision of the Eurasian and Arabian plates during the mid to late Miocene eliminated this oceanic connection, and sirenians in the Mediterranean evolved along separate paths from lineages along the coasts of Africa, Arabia, and southwestern and southeastern Asia. Understanding the identity and potential diversity of sirenians from Qatar, which was located along the southern margin of the Zagros Basin on the Arabian Plate during the early Miocene, would bear on the evolution of sirenian assemblages in the Tethys in the Neogene.

Elsewhere in the world, sea cow assemblages show convergent features in niche partitioning that evolved iteratively, since the late Oligocene (*Vélez-Juarbe, Domning & Pyenson, 2012*). It remains unclear if the isolated rib from northern Qatar represents a relictual lineage of early Miocene dugongids in Qatar, or a different lineage of Dugongidae from elsewhere in the Tethys, or one of the first members of *Dugong* in the Gulf. More data from sirenian fossils from marine deposits in Qatar and the Gulf Region would similarly confirm whether Miocene and Pleistocene sea cows there were more similar to the multispecies assemblages found elsewhere in the world from the Oligocene-Pliocene, or more similar to the singular lineages observed for extant sirenians today.

## CONCLUSIONS

We identified an isolated, incomplete dugongid second rib from the Futaisi Member of the Pleistocene Fuwayrit Formation near the town of Al Ruwais, in northern Qatar. The exposed section of the Futaisi Member in northern Qatar is approximately 125 ka in age. This sirenian occurrence is among the few pre-Holocene marine mammal records surrounding the Gulf. Among described sirenians, the morphology of this rib is closest to *Dugong*, but we cannot exclude the possibility that it belongs to an extinct taxon, either one already present in Qatar since the early Miocene or elsewhere in the world. The presence of Pleistocene dugongids in the Gulf Region suggests that seagrass communities were already present at this time in the region (*Vélez-Juarbe, 2014*), subsequent to the pre-Pliocene orogeny of the Zagros mountains and the formation of the Gulf. More data from the fossil record of sirenians in Qatar might bear on the question of whether different (and potentially multiple) lineages of sirenians inhabited the Gulf Region in the geologic past.

## ACKNOWLEDGEMENTS

We thank the Editor (K. De Baets) for comments and suggestions, as well as reviews from D. P. Domning, J. Gelfo, and an anonymous reviewer for edits that improved this manuscript. We thank A. H. Pyenson for his assistance with R for analyzing the comparative fossil and extant sirenian data, and his help drafting Fig. 3. We also thank M. R. McGowen, D. Lunde, and J. J. Ososky for access to modern comparative material, and D. J. Bohaska and A. J. Millhouse for access to fossil material. We thank D. P. Domning for his help clarifying the taxonomic identity of prorastomid sirenian material in

USNM collections. B. Felts was instrumental in pursuing much needed bibliographic references.

### Funding

Fieldwork for Nicholas D. Pyenson was supported by the Kellogg Fund and the MacMillan Fund from the Department of Paleobiology in the Smithsonian Institution's National Museum of Natural History. Fieldwork for Christopher D. Marshall was supported by Qatar National Research Fund (NPRP No. 11S-0102-180177). The funders had no role in study design, data collection and analysis, decision to publish, or preparation of the manuscript.

### Grant Disclosures

The following grant information was disclosed by the authors:
Smithsonian Institution's National Museum of Natural History.
Qatar National Research Fund: 11S-0102-180177.

### Competing Interests

Nicholas D. Pyenson is an Academic Editor for PeerJ and an Advisory Board Member for Arab Youth Climate Movement Qatar. Ismail Al Shaikh is employed by ExxonMobil Research Qatar.

### Author Contributions

- Nicholas D. Pyenson conceived and designed the experiments, performed the experiments, analyzed the data, prepared figures and/or tables, authored or reviewed drafts of the article, and approved the final draft.
- Mehsin Al-Ansi conceived and designed the experiments, performed the experiments, analyzed the data, authored or reviewed drafts of the article, and approved the final draft.
- Clare M. Fieseler conceived and designed the experiments, performed the experiments, analyzed the data, prepared figures and/or tables, authored or reviewed drafts of the article, and approved the final draft.
- Khalid Hassan Al Jaber conceived and designed the experiments, performed the experiments, analyzed the data, authored or reviewed drafts of the article, and approved the final draft.
- Katherine D. Klim conceived and designed the experiments, performed the experiments, analyzed the data, prepared figures and/or tables, authored or reviewed drafts of the article, and approved the final draft.
- Jacques LeBlanc conceived and designed the experiments, performed the experiments, analyzed the data, authored or reviewed drafts of the article, and approved the final draft.
- Ahmad Mujthaba Dheen Mohamed performed the experiments, analyzed the data, authored or reviewed drafts of the article, and approved the final draft.
- Ismail Al-Shaikh conceived and designed the experiments, performed the experiments, analyzed the data, authored or reviewed drafts of the article, and approved the final draft.

- Christopher D. Marshall conceived and designed the experiments, performed the experiments, analyzed the data, prepared figures and/or tables, authored or reviewed drafts of the article, and approved the final draft.

## Field Study Permissions

The following information was supplied relating to field study approvals (*i.e.*, approving body and any reference numbers):

Field collection was authorized by the National Museum of Qatar, Qatar Museums.

## Data Availability

The comparative dataset of first and second rib measurements from fossil and extant sirenians is available in the Supplemental File.

The specimens listed in the Materials and Methods that were used in the comparative dataset are deposited at the following institutions: the National Museum of Qatar, Qatar Museum Authority, Doha, State of Qatar; the University of Michigan Museum of Paleontology and the Geological Survey of Pakistan-University of Michigan collection at the University of Michigan Museum of Paleontology, Ann Arbor, Michigan, USA; and the Departments of Paleobiology and Vertebrate Zoology, National Museum of Natural History, Smithsonian Institution, Washington, District of Columbia, USA. See Materials and Methods and the Supplemental File for museum abbreviations.

## Supplemental Information

Supplemental information for this article can be found online at http://dx.doi.org/10.7717/peerj.14075#supplemental-information.

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
