# Peer review of "Fossil Sirenia from the Pleistocene of Qatar: new questions about the antiquity of sea cows in the Gulf Region"

_PeerJ, doi:10.7717/peerj.14075_

## Round 0.1 · original submission · Major Revisions

This manuscript reports the first remains of Dugongidae (likely Dugong) from the Pleistocene of the Fuwayrit Formation in northern Qatar which has important implications for the presence of sirenians post-Pliocene in the region and comparisons with other regions during this time. I would like to see it published, but there are some points which need to be addressed before publication:

Taxonomic justification: The available material concerns an isolated rib which is difficult to assign precisely. I agree with reviewer 2 that a more detailed justification to why this is the rib belongs to sirenian is necessary particularly for paleontologists not so familiar with such material. Ideally quantitative (geometric) morphometric assessment or at least visual or descriptive comparison with ribs of similarly sized terrestrial mammals, cetacean, pinniped, etc. would help in this endeavor. I agree with the reviewers 1 and 3 and your own assessment that an identification as Dugong dugon based on geographic and stratigraphic context is likely but not certain. In this context, it would like to see a more detailed justification (ideally in the form of a traditional paleontological section) as well as a comprehensive comparison with other fossil sirenians particularly those from the Indian ocean (compare reviewer 3).

Stratigraphic terminology: please make sure formal and informal units are correctly addressed and follow the latest edition of the international stratigraphic chart (see reviewer 2)

Implications of this fossil discovery: I feel that highlighting the importance of this find and potentially future finds of Dugongidae from the Pleistocene of the Gulf region for the differences/similarities between the Atlantic ocean/Mediterranean and Indian ocean sirenian faunas would broaden the scope of your study (compare Reviewer 1 and 3).

Typographical and formatting issues: please address the issues raised by reviewers 1, 2 and 3.

I consider most of these points to be crucial, but minor. However, as some of the raised aspects are more comprehensive and taxonomical in nature - and I am not an expert on Dugongidae – I plan to send the revised version for re-review.

I look forward to receiving the revised manuscript.

Please make sure these as well as all other points raised by the reviewers (including those in annotated pdfs) are addressed in the revision.

·

Basic reporting

This paper is professionally written. I have made numerous suggestions and corrections of typographical errors and punctuation on the Word copy of the MS. that I am sending via e-mail, but these are all minor. I have suggested citing one additional paper (Samonds et al., 2019) on which I am a coauthor; I think it sheds new light on the differences/similarities between Mediterranean and Indian Ocean sirenian faunas, and should lead the authors to update their discussion of this topic.

Experimental design

The very rich and detailed coverage of the geological, stratigraphic, and geochronological context of the sirenian rib will be of great use to future workers at the site and beyond; and the discussion of other occurrences of fossil sirenians in the wider region will be helpful in introducing local earth scientists to the sirenian fossil record.

Validity of the findings

The key specimen being reported is unfortunately not well preserved and lacks key diagnostic features; so the authors are not able to determine with certainty its taxon (Dugong dugon or an extinct taxon) or even element (rib 1 or 2). Their identification is most probably correct (not least because it is the species that would be expected, given its locality and age); but it could certainly use corroboration by collection of better material. The major value of this paper thus lies in calling attention to the locality and putting it on record so that future collectors can confirm the Pleistocene occurrence there of D. dugon.

Additional comments

No comment.

·

Basic reporting

The article is well written, the English is clear and unambiguous. There are few technical things to correct about the temporal use of some ages. For example, line 69 should be early Miocene and not Early Miocene, since it is an informal unit. The formal units to divide the Miocene Epoch in ages are Aquitanian, Burdigalian, Langhian, Serravallian, Tortonian and Messinian. The same should be done in lines 75, 274, 278, 283, 292, 297 and 310. Check the last edition of the international stratigraphic chart at https://stratigraphy.org/.

The introduction could benefit of a more detail comment about dugongs fossil record which highlight the possible paleontological importance of the present finding. There is an important description of present dugongs distribution and general comments of Qatar fossil record in the Cenozoic, but nothing about other worldwide fossil record of them. Literature references are fine, but it could be improved with a paragraph of dugongs fossil record at the introductory section.

I prefer a more traditionally structure for a paleontology paper. Perhaps a classic section of “Systematic Paleontology” added after Materials and Methods would help the readers. I suggest to put there not only the basic taxonomy inferences, but also a short detail of geographic and stratigraphic distribution, description and discussion sections.

Experimental design

The taxonomic identification is based in a very short description of what were alleged as sirenian diagnostic characters. This is a bit soft for me and rise doubts about the taxonomic assignment, since they only have a part of a broken rib. I’m not a specialist in sirenian ribs and may be the description and photograph are self-evident in order to identified it. But I think than even in this situation it could be more accurate for reinforce the sirenian assignation, to shows comparisons with other mammalian ribs. This could be done with the addition of a statistical analysis, geometric morphometric or simply by photographs in order to compare QM.2021.0504 with ribs of terrestrial mammals, cetacean, pinniped, etc. indicating the way to discard other taxonomic possibilities. Also the other fossil dugongs from Qatar should be in my opinion incorporated to the main text as part of the discussion.

Validity of the findings

In my opinion is it completely possible that this rib is as author said from a dugong. But, in my humble opinion I think they should re-structure more precisely how to justify it as I mentioned before.

Reviewer 3 ·

Basic reporting

This paper reports an isolated fragment of a rib of a dugongid from Qatar, in the Persian Gulf. The paper is relatively well structured. I detected some grammatical errors and I suggest a throughout revision.

Experimental design

No comment

Validity of the findings

Almost certainly the fossil reported belongs to Dugong or a very close relative. Considering the age of the material (inferred to be Pleistocene), and that the diversified dugongid communities known from the Miocene are mostly gone by the Pliocene, I believe that the discussion should have focused more on the implications for the distribution of Dugong and paleoenvironmental implications. I agree that, with the material at hand, an identity as another species of dugongid cannot be completely discarded, and should be discussed: however, the circumstantial evidence against it should be mentioned, and more emphasis given to those aspects mentioned above.

Additional comments

One unfortunate trait of the study is that all comparisons were made with Atlantic (and one North Pacific) sirenians, unlikely candidates for the identity of the sirenian rib described. I think that it is unlikely that a more specific identity could be reached considering the paucity of the material available, but maybe it would be more interesting to compare it to other dugongids from the Indian Ocean, like Kutchisiren or Domningia. However, considering the age and locality of the fossil, I totally agree that it is way more likely that the material is attributable to Dugong, although I also agree that the identity as an undetermined taxon can not be discarded.

On line 91, I believe the currently accepted name for this species is Trichechus manatus.

---

## Round 0.2 · accepted · Accept

Thank you for taking the time to thoroughly address the reviewer comments. I agree with the recommendations by the reviewers that the paper can be accepted pending some minor typos (compare reviewer 1) and one species name (compare reviewer 3) are corrected during the proofing phase. As succinctly described by reviewer 3, the comparisons with other sirenians has markedly improved - both from a qualitative and quantitative side - which provide a more solid base for its identification of the material as well as its limitations.

·

Basic reporting

I have made a few additional minor edits and corrections of typos (with tracked changes) on a file I am submitting separately by e-mail. I am satisfied that the paper should be published in its present form, with no further review required.

Experimental design

No comment.

Validity of the findings

No comment.

Additional comments

No comment.

**Staff Note** The reviewer's comments are attached as a PDF and we will forward the Word document separately.

·

Basic reporting

The article is professionall and well written, the English is clear and unambiguous. The literature references were updated. The authors have harmonized the Age mentions to conform with the International Stratigraphy Commission throughout the revised manuscript text, and now is much better. The authors also have added text to both the Introduction and Results & Discussion to provide more background on the Mediterranean and Indian Ocean sirenian faunas and the implication of the Qatari finding, which clearly improves the manuscriopt. They also added a Systematic Paleontology section with basic taxonomy as previously suggested. On the contrary they kept the stratigraphy and morphological description sections at their current length, which is fine with me.

Experimental design

Previously, I mention that the taxonomic identification is based in a very short description of what were alleged as sirenian diagnostic characters of a broken rib. As I said, I’m not a specialist in sirenian ribs and may be the description and photograph are self-evident in order to identified it. I the other reviewers agree with the authors is ok for me.

Validity of the findings

No comments.

Additional comments

No comments.

Reviewer 3 ·

Basic reporting

This is the second review I was invited to do about this paper reporting an isolated rib fragment from Qatar. Compared with the previous version, this new one is much improved. The comparison with other sirenians, both in qualitative and quantitative terms, is much more thorough and gives a much more solid base for the case advocated by the authors about the identity proposed and its limitations. I was particularly pleased by the detailed morphological comparisons and descriptions. I also think most of the points I rased in my previous revision were satisfactorily answered.

Experimental design

No comment.

Validity of the findings

I think that the paper is a significant contribution to sirenian distribution and evolution in the eastern Tethys region and I am pleased to recommend it for publication.

Additional comments

On page 10, line 119, the correct name of the species would be Trichechus inunguis. That is the only reason that I would not recommend acceptance as it is.